# Dermal Papilla Cells: From Basic Research to Translational Applications

**DOI:** 10.3390/biology13100842

**Published:** 2024-10-20

**Authors:** He-Li Zhang, Xi-Xi Qiu, Xin-Hua Liao

**Affiliations:** 1School of Environmental and Chemical Engineering, Shanghai University, Shanghai 200444, China; helizhang607@foxmail.com; 2School of Life Sciences, Shanghai University, Shanghai 200444, China; qiuxixi001@163.com

**Keywords:** dermal papillae, mesenchymal stem cells, cell therapy, hair cycle, hair loss

## Abstract

The dermal papilla (DP) is a specialized mesenchymal compartment that forms at the base of the hair follicle. It acts as a signaling center that regulates hair growth, shape, size, and color. Dermal papilla cells (DPCs) exhibit stem cell properties and show significant potential for use in cell therapy. They can be induced into other functional cells, regenerate skin tissues and new hair follicles when combined with epithelial stem cells, and promote hair growth and wound healing when administered on the skin. This review summarizes the functions of DPs in the hair follicle, shows how DPCs could be used in cell therapy, and discusses the challenges and recent advances in the field, from basic research to translational applications.

## 1. Introduction

Hair is generated by the skin appendage, the hair follicle, through interactions between epithelial and mesenchymal cells [1]. At the base of the hair follicle lies a structure called the dermal papilla (DP), which is formed by specialized mesenchymal cells during development and continues to release signaling molecules to the epithelial cells after hair follicle maturation, controlling hair growth, shape, size, and color [2,3,4,5]. Dermal papilla cells (DPCs) are a unique type of mesenchymal cell that can be cultured and expanded in vitro [6], and increasing evidence suggests their potential in cell therapy. This article reviews the functions of DPCs in hair follicle development and hair growth, provides a detailed description of their potential applications as specialized mesenchymal stem cells (MSCs), discusses recent research progress, and critically analyzes the challenges and limitations in translating these findings into clinical practice, highlighting the areas in which breakthroughs are needed to overcome these barriers.

## 2. Formation of DP and Its Function During Hair Follicle Morphogenesis

Basic research in mice reveals that hair follicle development during embryogenesis involves a series of reciprocal interactions between epithelial and mesenchymal cells [7]. Initially, dermal mesenchymal cells release signals to the epithelial cells, inducing epithelium thickening to form the placode. Conversely, the placode signals to the underlying dermal mesenchymal cells, inducing cell proliferation to form dermal condensates (DC) [8]. The reciprocal signaling interactions induce invagination and a downward extension of the basal cells of the placode, which ultimately envelop the DC. This process results in the maturation of the DP from the DC and ultimately leads to the maturation of the hair follicle [4]. During hair follicle formation, inner basal cells differentiate into the inner root sheath (IRS), which encapsulates the future hair shaft [9], while outer basal cells differentiate into the outer root sheath (ORS) [10]. The bulge region within the ORS contains hair follicle stem cells (HFSCs) and melanocyte stem cells [11]. The ORS cells differentiate into highly proliferative matrix cells at the bulb region, which surround the DP [12,13]. Melanocyte stem cells migrate and differentiate alongside HFSCs, and mature melanocytes reside within the matrix cells [14,15].

Key signaling pathways, such as the wingless/integrated (WNT), sonic hedgehog (SHH), and bone morphogenetic protein (BMP) pathways, are crucial in this process (Figure 1).

The WNT signal is recognized as the “first dermal signal” in HF morphogenesis [16], which precedes and is essential for the localized expression of regulatory genes and the initiation of HF placode formation from a uniform single layer of the basal epidermis [16,17]. The WNT activity in the placode epithelium promotes dermal cell proliferation and induces dermal condensate (DC) formation [18,19]. The β-catenin protein is uniformly activated in the upper dermis before its activation in the placode epithelium, followed by its activation in the underlying mesenchyme and DC [20,21]. The feedback from the dermal WNT then promotes the full induction of the placode signals [19,21].

SHH pathway, which acts downstream of the WNT signaling, is expressed in the epithelial cells and the DC, playing a crucial role in signaling from the epithelium to both the epithelial and mesenchymal cells [22,23]. SHH promotes the proliferation of the HF epithelium and its subsequent downward growth, as well as the formation of the DC [24]. While the induction of HFs is independent of SHH, its role in the ingrowth of the epidermis and the subsequent morphogenesis of the hair shaft is essential [23,25]. SHH signaling regulates specific DP signatures and maintains a reciprocal SHH–Noggin (BMP inhibitor) signaling loop to drive hair follicle morphogenesis [23,26].

BMP signaling acts in an inhibitory manner during HF morphogenesis. The ligands BMP2 and BMP4 are enriched in the placode and DC [27,28]. BMP signaling, along with its reporters and antagonists, exhibits a strict spatiotemporal expression pattern to properly regulate cell proliferation and differentiation in the epidermis and HFs [29,30]. The BMP receptor BMPR1A, expressed in HFs, is crucial for the differentiation of IRS and hair shaft progenitor cells [31,32]. Without BMPR1A activation, the terminal differentiation factor GATA-3 is downregulated, compromising its regulatory control over IRS differentiation [33]. BMP signaling is essential for HFSC quiescence and the promotion of TAC differentiation along different lineages during the hair cycle [34,35].

It is noteworthy that dermal fibroblasts and thus, DP, originate from different precursor cells in different parts of the body. Lineage tracing assays in mice show that dermal fibroblasts in the dorsal and ventral trunk originate from the somite, whereas dermal fibroblasts in the head and face originate from the neural crest [36,37]. The neural crest cells in birds and mice also migrate to the skin and give rise to melanocytes during embryogenesis [37,38]. Ablation of the WNT signaling in the mouse dermis leads to ectopic cartilage formation in the craniofacial and ventral trunk regions at the expense of the dermal and bone lineages [39]. Dermo-1/twist2 (twist basic helix-loop-helix transcription factor 2), a downstream transcription factor of WNT signaling, may mediate the function of WNT signaling in dermal cell development [39,40]. Dermo-1 knockout mice showed radically thin skin and sparse hair [41]. The forced expression of chicken Dermo-1 induces a dense dermis, feathers, and scales [42]. These findings highlight the significance of Dermo-1 in dermis maturation from different embryonic origins, which is crucial for hair follicle development and patterning.

In summary, DC/DP serves as a signaling center for hair follicle development, driving epithelial proliferation, placode formation, and downward growth of the hair follicle. This process involves diverse interactions between epithelial and mesenchymal cells, ultimately leading to the formation of different hair follicle lineage cells and the maturation of hair follicles [43,44].

## 3. DP Regulation in Mature Hair Follicles

### 3.1. DP Regulates Hair Growth/Cycle

After the maturation of the hair follicle, hair undergoes cyclical growth. The mature DP continues to serve as the signaling center for the hair follicles, finely regulating the quiescence, activation, proliferation, and differentiation of hair follicle epithelial cells, thereby controlling the hair cycle (Figure 1) [45,46]. During the growth phase (anagen), hair germ cells, which are derived directly from bulge HFSCs, proliferate to form transient amplifying matrix cells (TACs) [47]. These TACs surround the DPCs and receive signals from them, ultimately differentiating into hair shafts. In the regression phase (catagen), matrix cell proliferation ceases, and skin cell apoptosis occurs, leading to the degeneration of two-thirds of the lower follicle, while DPCs remain intact and migrate upwards along the epithelium [48]. In the resting phase (telogen), epithelial cells of the hair follicle remain in a quiescent state, with the DP located at the tip of the hair follicle, directly in contact with hair germ cells [49,50]. When quiescent hair germ cells receive activating signals from the DP, they rapidly proliferate into progenitor cells, which then differentiate into various mature cells, initiating a new hair growth cycle [50]. It should be noted that in mice, the growth cycle of all hairs on the back is synchronized until the second anagen, whereas in the adult human scalp, individual hairs maintain their own growth cycles. At any given time, approximately 90% of human scalp hair follicles are in the anagen phase, 1% are in the catagen phase, and 9% are in the telogen phase [51].

DPCs not only secrete factors such as WNTs [52], R-Spondins [53], transforming growth factor-β2 (TGF-β2) [54], hepatocyte growth factor (HGF) [55], and insulin-like growth factor 1 (IGF1) [56] to activate the WNT pathway and promote the proliferation of HFSCs [57], but also express the growth factors fibroblast growth factor 7 (FGF7) and FGF10 [58] to enhance the SHH signal in the hair follicle epithelial cells (Figure 1). The SHH signal acts downstream of WNT to stimulate the neighboring epithelial cells of the hair follicle [59], thereby accelerating hair growth [60].

Among these factors, DP-expressed R-Spondins potentiate WNT signaling to stimulate the proliferation of both HFSCs and epithelial progenitors during HF regeneration [61]. TGF-β2 activates the Smad2/3 pathway in HF stem cells, and antagonizes BMP signaling in HFSCs, thereby promoting the transition from the telogen to the anagen phase of the hair cycle [62]. HGF stimulated WNT/β-catenin activity in the hair matrix by upregulating WNT6 and WNT10B and inhibiting secreted frizzled-related protein 1 (SFRP1) in DP [55]. IGF-1 acts as an anti-apoptotic survival factor to inhibit cell death during the catagen phase of the hair cycle [63], as well as stimulates cell proliferation to promote hair growth [64]. The FGF and WNT signaling pathways interact in the DP to regulate the hair cycle by orchestrating the expression of WNT agonists (R-Spondins) and antagonists (Dickkopf-related protein 2 (DKK2) and Notum) [65]. Furthermore, DPCs express high levels of BMP pathway-related proteins such as BMP4, BMP6, BMP7, BMP receptor BMPR1A, and BMP inhibitor Noggin [66,67], which are involved in the complex mechanisms governing the transition of the hair follicle from the growth phase to the resting phase [68]. Noggin, with a 10–15 times higher affinity for BMP4 compared to that of BMPR-1A, likely prevents BMP4 interaction with BMPR-IA expressed in the secondary germ of the telogen hair follicles, thereby stimulating anagen initiation [69]. 

Previous studies have shown that the selective ablation of certain DPCs in mice delays the growth phase of the hair follicles [2]. Furthermore, the complete laser ablation of DPCs prevents hair follicles from entering the growth phase [70]. Under chronic stress, the hormone corticosterone, derived from the adrenal gland, inhibits growth arrest-specific gene 6 (GAS6) in DPCs, governing HFSCs’ quiescence and leading to hair loss [71]. Previous studies also suggest that the androgen receptor (AR), which is specifically expressed in DPCs in the skin, may be involved in androgenic alopecia [72].

### 3.2. DP Regulates Hair Properties

In mammals, melanocytes determine the pigment deposition in the skin and hair [73]. The expression of melanin from melanocytes in the back skin of mice strictly follows the hair growth cycle, peaking in the late growth phase [74]. Previous studies have reported that DPCs regulate melanocytes via the WNT signaling pathway [75]. The plucking of hair from the backs of mice leads to the upregulation of the secretory factor endothelin 3 (EDN3) in the DPCs [76], and the deletion of EDN3 leads to coat white spotting in mice [76,77]. It has also been found that AGOUTI (also known as ASIP, agouti-signaling protein), specifically expressed in DPCs, competes with α-melanocyte-stimulating hormone (α-MSH) for binding to the melanocortin 1 receptor (MC1R), promoting the production of red/yellow pheomelanin instead of black/brown eumelanin [78,79,80]. Transient expression of AGOUTI in mice causes the production of a yellow-striped band near the tip of the black hair shaft. The serine protease CORIN is specifically expressed in DPCs during the growth phase and acts downstream of AGOUTI, inhibiting its pathway and balancing the production of pheomelanin and eumelanin [81]. The sex determining region Y-box 2 (SOX2) in DP regulates hair pigmentation through the upregulation of AGOUTI and the downregulation of CORIN [82,83].

The types of hair on the backs of mice include guard, awl, auchene, and zigzag, which form at different stages of development [84,85]. Guard hairs are the longest and smoothest, emerging the earliest during embryonic day 14.5; awl hairs are larger and blunter, appearing from embryonic day 16.5; auchene hairs, characterized by their bent middle, are the least abundant and also appear from embryonic day 16.5; zigzag hairs have 3–4 bends and are the most abundant, appearing from embryonic day 18.5 [84,85,86]. DPCs determine the different hair types [6,27]. SOX2 is expressed only in guard, awl, and auchene DP but not in zigzag DP. The removal of SOX2+ cells from DP results in the reconstitution of hair with only zigzag hairs, without the expression of awl and auchene hairs [87]. Sox18 knockout mice display a mild coat defect, with a reduced proportion of zigzag hairs and an increased proportion of auchene hairs [88,89]. Our recent study has demonstrated that the transcription factor early B cell factor-1 (EBF1), which is prominently expressed in DP, also plays a role in determining the type and length of hair. Specifically, the conditional knockout of EBF1 in DPCs leads to a reduction in hair length, a decrease in the amount of awl hair, and an increase in the amount of zigzag hair [90].

The number of DPCs within each hair follicle correlates with and determines the size and shape of the hair. Anatomical examination of hair follicles at postnatal day 11 reveals significant variations in the number of DPCs among the four types of follicles. As one type of hair transitions into larger and longer hairs, the number of DPCs significantly increases [2,91,92]. The selective ablation of DPCs in vivo using genetic mice models results in smaller hairs [2]. The genome sequencing of dogs with varying hair morphology has linked DP-expressed R-Spondin 2 (RSPO2) to properties such as the length and curliness of dog hair [93]. Scientists have discovered that the Hoxc gene family (HOXCs), expressed in mice DPCs, influences the activation of the downstream WNT signaling pathway, thereby affecting the length of hair across different parts of the mouse body [94].

In summary, the number and gene expression of DPCs correlate with or even determine hair properties such as hair color, size, type, and morphology, suggesting that they play an important role in hair formation.

## 4. Technological Advances in DP Basic Research and Establishment of Genetically Modified Mice as a Tool for Studying Functions of DP Genes

In the past, studies on the regulation of hair growth primarily focused on HFSCs, thanks to the creation of various genetically modified mice, such as K14-Cre, for specific gene knockout in skin epithelium [95]; K14-rtTA, for specific gene overexpression in skin epithelium [96]; and K19-CreERT, for specific gene knockout in HFSCs [97]. Due to the important role of DP in hair regulation, there has been increasing research on the functions of genes in DP. The mice reported as tools for gene knockout in DP include Tbx18-Cre [82,98], Corin-Cre [5], Sox2-CreER [99] and CD133-CreERT 2 [100,101], Hey2-CreERT2 [102], Lepr-Cre [90], and LeprB-Cre [103]. The agouti gene, which regulates hair coat color, is confined to expression in the DP after birth, but is widely expressed in dermal fibroblast precursors during embryonic development. Consequently, Agouti-Cre mice will induce gene knockout in both the DP and all dermal fibroblasts [104]. Tbx18-Cre targets precursor cells in DP [82,98], Corin-Cre targets DPCs after hair follicle formation [5], Sox2-CreER targets DPCs in guard, awl, and auchene hair follicles [99], and CD133-CreERT2 targets some DPCs [100,101]. 

These Cre mice help to reveal the regulatory functions of DP genes such as SOX2 [82,83,87,99,105], β-catenin [5,101], Fgfr1 and Fgfr2 [65], and EBF1 [90] in regards to hair properties. For example, the ablation or activation of β-Catenin, specifically in mice DP reveals, that β-Catenin in DP functions by switching the hair color from yellow to black, as well as by promoting postnatal hair growth [5,101]. Mice with Fgfr1/2 double knockout, specifically in DP, exhibit extremely long hairs [65].

One reason scientists continue to search for Cre mice is that existing mice lack specificity. Since DPCs originate from mesenchymal cells in the dermis [60,106], the Cre recombinase in these mice is more or less expressed in other dermal cells similar to DPCs, such as DS cells connected to DP and a small number of other dermal fibroblasts. Due to the similarity between DP and DS [91,107], it is difficult to achieve DP-specific expression using a single gene promoter to control Cre expression. However, using two promoters, such as those active in genes expressed in both DP and DS, along with those active specifically near the DP niche (e.g., Lef1), may eventually achieve DP-specific expression using both the Cre-LoxP and Dre-Rox systems [108]. It is worth mentioning that CreER mice can also be used to trace the lineage of DPCs in vivo [102], revealing whether they generate DS cells, whether they participate in dermal fibroblast generation during wound healing, and whether they participate in new DS and DP generation during hair follicle neogenesis after injury [109]. Obtaining definitive conclusions still requires the acquisition of dual-labeled, DP-specific CreER mice.

Another reason may be the significant cost required to transport these mice between different countries. In addition to Cre mice, establishing rTTA mice for gene overexpression specifically in DP will further facilitate the study of gene functions in DP [101].

## 5. Stemness and Potential Applications of DPCs in Cell Therapy

As the signaling centers for regulating hair follicle development and hair growth, DPCs express high levels of numerous growth factors. DPCs are particularly notable for their stem cell properties. This unique property makes them very promising for the development of cell products for cell therapy.

### 5.1. Reprogramming of DPCs into iPSCs

Compared with other somatic cells, DPCs express various stem cell factors and are more easily induced into pluripotent stem cells (iPSCs) [110]. Typically, the induction of mice embryonic or adult fibroblasts into iPSCs requires the addition of four transcription factors, i.e., octamer-binding transcription factor 4 (OCT4), SOX2, Krüppel-like factor 4 (KLF4), and and cellular myelocytomatosis oncogene (C-MYC) [111]. However, DPCs themselves express SOX2, C-MYC, and KLF4, so their induction into iPSCs only requires the addition of OCT4 and KLF4, or even just OCT4 [112,113]. Under the same conditions, the efficiency of reprogramming DPCs into iPSCs is about three times higher than that of fibroblasts [106].

### 5.2. DPCs Are a Main Source of SKPs

Skin-derived precursors (SKPs) are multipotent progenitor cells in the skin that are capable of self-renewal and display the characteristics of multipotent neural stem cells [114]. Freda D. Miller’s laboratory initially dissociated mice dermal tissues into single cells and cultured them in a medium containing epidermal growth factor (EGF) or FGF, where some cells formed floating spheres and proliferated during passage [115]. These cells differentiate along neuronal and glial cell lineages, while also expressing markers of MSCs, differentiating into smooth muscle cells, adipocytes, and other types of cells. Subsequently, the same laboratory found that SKPs derived from adult dermis possess characteristics similar to those of embryonic neural crest stem cells (NCSCs), with DP providing a niche for SKPs. Cell lineage tracing has shown that both mice hair and whisker follicle DP contain NCSCs, and whisker pad SKPs also originate from NCSCs, suggesting that DPCs are likely a main source of SKPs [116]. Further studies have shown that SOX2+ cells are distributed in DP and the lower parts of the dermal sheath (DS), and SKPs and SOX2+ cells cultured in vitro exhibit similar gene expression profiles and functions, i.e., both can reconstitute hair follicles when combined with epithelial cells from neonatal mice, differentiate into dermal fibroblasts, and regenerate DP and DS [117].

SKPs can also differentiate into adipocytes, hepatocytes, keratinocytes, melanocytes, and other cell types in vitro [118,119]. Steinbach et al. demonstrated that rat SKPs can be easily differentiated into functional vascular smooth muscle cells in vitro, laying the foundation for SKPs’ application in the treatment of ischemic diseases [120]. When transplanted into a normal or injured hippocampus, mice SKPs can survive and maintain their neural characteristics for at least five weeks [121]. When transplanted into the dysmyelinated brain, naive rodents or human SKPs can generate Schwann cells and myelinate central nervous system (CNS) axons [122,123]. SKPs can be used to treat spinal cord injuries by generating Schwann cells [123]. It has also been reported that SKPs transplanted into the aganglionic colon of pigs survive and express neuroglial differentiation markers [124,125]. SKPs differentiate into skeletogenic cell types in vivo to contribute to bone repair [126]. Human SKP-derived corneal endothelial cell-like cells transplanted into a monkey model of corneal endothelial dysfunction maintain functionality for two years [127,128]. 

DPCs can differentiate into adipocytes and osteoblasts using specific differentiation mediums, although whether DPCs have the same differentiation potential as SKPs remains to be verified [129]. Nevertheless, the fact that DPCs are a main source of SKPs indicates there is enormous potential for SKP-based cell therapies.

It is worth noting that Nestin-expressing hair follicle-accessible pluripotent stem cells (HAP) [130] are likely terminal Schwann cells that wrap around the hair follicle isthmus [131]. Human hair follicle-derived mesenchymal stem cells (HF-MSC) are derived from the DP and DS cells grown in a culture medium from isolated hair follicles [132]. HAP and HF-MSC both show characteristics of SKPs and are likely sources of SKPs, although their isolation methods are different.

### 5.3. DPCs and Their Derived Exosomes Promote Wound Healing

It has been reported that MSCs and their extracellular vesicles have significant potential to promote tissue repair and regeneration [133]. DPCs, as a special type of MSC, can differentiate into dermal fibroblasts during skin regeneration and wound healing; additionally, various growth factors and cytokines, either from DPCs or their derived exosomes, can promote angiogenesis, reduce inflammation, and facilitate wound healing [102,117,134,135,136,137].

### 5.4. DPCs Induce Hair Follicle Neogenesis In Vivo

Removal of the lower third of the whisker follicle leads to the loss of the ability to regenerate whiskers when the follicle is transplanted into ear skin. However, freshly isolated or culture-expanded DPCs implanted into the bases of rodent whisker follicles can restore the ability to grow whiskers [138,139]. Implanting DPCs between the epidermis and dermis of the hairless footpads in rats induces de novo hair follicle formation and hair growth [140]. Similarly, implanting DPCs between the epidermis and dermis of the human foreskin can induce hair growth on hairless skin [141]. Implanting DPCs into wounds on rat ears and backs has been found to aid in wound healing and induce new hair follicles [134,138]. Mixing DPCs with epithelial stem cells and transplanting them onto the backs of nude mice can reconstitute hairy skin [67,142,143,144].

### 5.5. DPCs and Their Derived Exosomes Promote Hair Growth

DPCs secrete various bioactive molecules such as growth factors and cytokines, which promote the proliferation and differentiation of HFSCs. The subcutaneous injection of mice DPCs can induce hair follicles from the resting phase into the growth phase and delay the onset of the regression phase, thereby promoting hair growth [145,146,147]. Recent studies have identified similar bioactive molecules in exosomes from cultured dermal papilla cells; injecting cell-free exosomes subcutaneously can also improve the scalp microenvironment, activate HFSCs, and promote hair growth [145,146,148,149,150]. The application of DPC-derived exosomes onto mouse full-thickness skin excisional wounds also promotes hair follicle neogenesis [137]. Therefore, the use of DPCs and their exosomes provides a potential therapeutic strategy for hair loss.

## 6. Strategies and Challenges of DPCs-Based Cell Therapy for Treating Alopecia

Human hair performs a wide range of functions, including thermoregulation, sensation of touch, and protection against ultraviolet radiation and mechanical damage [47,151,152]. Hair also reflects an individual’s health status, and hairstyles serve as an important method of self-expression. Loss of hair (alopecia) lowers an individual’s self-esteem and causes psychological distress [153].

Current treatments for alopecia do not meet the clinical needs of patients. For example, hair transplantation represents one of the most popular treatments for alopecia [154]. However, this method is constrained by the limited number of hair follicles available in the donor area and the cost of the treatment. Minoxidil and finasteride are the only two medications approved by the U.S. Food and Drug Administration for the treatment of alopecia, yet these treatments offer limited efficacy and cause significant side effects [155,156].

Based on their functions in inducing hair follicle neogenesis and promoting hair growth, DPCs can be applied in two potential strategies to treat alopecia: one involves reconstituting hair follicles, along with epithelial stem cells, and the other involves the subcutaneous injection of DPCs or their derived exosomes to promote hair growth (Figure 2). For the former strategy, the direct reconstitution of hairy skin on a patient’s scalp is impractical for the following reasons: (1) Current hair follicle reconstitution techniques in mice are based on hairy skin regeneration during wound healing, resulting in scar-like skin with impaired functionality [157]; (2) The density, size, and direction of hairs regenerated through spontaneous skin reconstitution from stem cells are uncontrollable [158]; (3) The reconstitution of colored hair also requires the inclusion of melanocyte stem cells in the reconstitution system [159]. An alternative approach involves reconstituting human follicles in immunodeficient large animals (e.g., pigs) and transplanting them to human scalps. However, hair follicles reconstituted in animals inevitably incorporate animal cells, such as vascular endothelial cells and dermal cells, raising questions about immune reactions and the survival of hair follicles after transplantation into human scalps. Furthermore, reconstituting hairy skin requires not only a large number of autologous DPCs but also melanocyte stem cells and HFSCs, which currently cannot be sufficiently maintained and expanded in vitro (Figure 2). 

To implement the latter strategy, it is essential to isolate and expand DPCs on a large scale. In the case of autologous DPCs, a small number of healthy DPCs must be expanded in vitro through numerous passages while retaining their hair follicle induction activities. It is important to consider the potential for immune rejection from allogeneic DPCs and their exosomes. While primary DPCs can be obtained in large quantities from animals, there are significant concerns regarding immune rejection and ethical barriers (Figure 2). A common limitation of cell therapy is the poor survival of transplanted cells due to the immune rejection of allogeneic cells and failure to integrate the transplanted cells into the recipient tissues/organs [160]. To reduce immune rejection, the deletion of MHC (major histocompatibility complex) expression in transplanted cells and pretreatment with immunosuppressive drugs are common measures [161,162]. Strategies to reduce apoptotic signaling and increase cell adhesion, such as pretreatment with cytokines, growth factors, and anti-apoptotic molecules, have been developed to improve the long-term survival of transplanted cells [163]. These pretreatments can likely be applied topically to the scalp when using DPCs to treat alopecia. In addition to traditional drug therapies and hair transplantation, new therapies have been developed for the treatment of alopecia. For example, platelet-rich plasma (PRP) [164], MSCs [165], and MSCs-derived exosomes [166] or conditioned media [167] are injected into the scalp to stimulate HFSCs. A common mechanism behind these therapies is that the injected substances contain growth factors. DPCs are unique in that they accumulate high concentrations of growth factors that activate HFSCs, such as R-Spondins, VEGF, FGF, HGF, etc. [102,168]. DPCs-based cell therapy would offer significant advantages over other therapies for the treatment of alopecia.

## 7. Technological Advances in DP Translational Research—Obtaining DPCs on a Large Scale

Obtaining sufficient numbers of DPCs is the main obstacle to DPCs-based cell therapy. To solve this problem, scientists have made significant technological advances, including the large-scale isolation of DPCs using surface protein antibodies, the improvement of culture conditions to allow massive expansion of DPCs in vitro, and a technique for inducing iPSCs into DPCs, which provides an unlimited source of DPCs. These technologies not only advance translational research, including using DPCs as a source of SKPs that differentiate into other functional cells such as neurons, reprogramming DPCs into iPSCs, and using DPCs to promote wound healing and hair follicle regeneration or hair growth to treat hair loss, but also provide DPCs for basic research, including studying the functions of DP genes in vitro (Figure 3).

### 7.1. Large Scale Isolation of DPCs

To study the functions of DP in hair follicle development, hair cycle regulation, and hair regeneration, or to apply DPCs in disease treatment, the isolation and in vitro culture of DPCs are indispensable. Traditional isolation methods include microdissection, which involves the use of a scalpel under a stereomicroscope to physically separate dozens of DPCs from a single hair follicle [169,170]. this method is time-consuming and is only suitable for large hair follicles, such as mice whisker follicles and human scalp hair follicles. it has been reported that dpcs can be sorted using antibodies against the surface protein cd133/prom1, followed by flow cytometry [171]. however, cd133 is a general stem/progenitor cell marker [172], not highly specific to dpcs, and is expressed in other cells such as ors cells and matrix cells in the skin [173]. there are reports of using genetically labeled dpcs in mice via fluorescent proteins under the control of the dp-specific lef1 promoter [66,67,174]. however, this method relies on the construction of transgenic animals and has not been reported in species other than mice. 

Our laboratory has identified a dp membrane protein, leptin receptor (lepr), and developed a more specific and efficient technology for functional dpc sorting on a large scale [168]. by analyzing the gene expression profile of dp cells using existing microarray and rna-seq databases, lepr was identified as the second most abundant surface protein in dp cells, exhibiting a higher level of specificity compared to that of the previously utilized cd133 marker [66,168,175]. the lepr antibody-based sorting method allows for the isolation of hundreds of thousands of dp cells in a few hours, a significant improvement in efficiency over that of traditional microdissection. Furthermore, LEPR+ cells sorted by this method retain their DP characteristics in vitro, as demonstrated by the expression of DP markers (such as alkaline phosphatase) and key secretory factors (such as RSPO1/2 and EDN3), as well as the successful reconstitution of hair follicles in nude mice when combined with epithelial stem cells [168].

### 7.2. Expansion of DPCs In Vitro

While undergoing culturing in a Petri dish, DPCs gradually lose their biological activity or stemness, especially the ability to induce hair follicle formation [176,177]. Generally, growth beyond the sixth passage of DPCs begins to slow down, accompanied by noticeable morphological changes such as the loss of aggregative growth [178,179]. In vivo, DPCs interact closely with epithelial cells during hair follicle development, with mature DPCs in direct contact with or even enveloped by epithelial cells [60]. The culturing of skin epithelial cells on feeder fibroblasts mimics the in vivo epithelial–mesenchymal cell interaction and has achieved long-term expansion [180]. Co-culturing DPCs with skin epithelial cells not only maintains the stemness of the epithelial cells [181], but also greatly extends the passage number of the DPCs. Similar effects are observed when DPCs are cultured using a skin epithelial cell-conditioned medium [147]. Under these conditions, DPCs can still maintain their hair follicle-inducing ability, even after 70 passages [182,183]. It has been reported that adding the growth factor FGF2 to DPCs can extend their passage number from 5 generations to over 30 generations [176].

Another successful approach is to culture DPCs into spheres using a three-dimensional suspension culture method, which facilitates cell–cell contact and enhances the ability of DPCs to induce hair follicle formation [141,176,184].

Additionally, immortalized DPCs have been established by introducing the SV40 large T antigen and the telomerase (TERT) gene [182,185,186]. These cell lines maintain the expression characteristics of DPCs, but whether or not they retain the ability to induce hair follicle formation is not mentioned.

### 7.3. Induction of iPSCs into DPCs

Another approach for obtaining a continuous supply of DPCs is through their induction from iPSCs. Scientists first induced iPSCs into neural crest stem cells (NCSCs) using recombinant Noggin and an inhibitor of activin/nodal and TGFβ signaling, and then further induced NCSCs into SKPs using a WNT signal activator in an SKP medium [187]. A similar trajectory was adopted to induce human iPSC differentiation into DP-like cells [188].

In another study, scientists differentiated iPSCs into induced mesenchymal cells (iMC) via a bone marrow stromal cell phenotype. Subsequently, using retinoic acid and a DP cell-activating culture medium, they reprogrammed a subset of LNGFR+ THY-1+ iMC to acquire DPC properties. When co-grafted with human keratinocytes in vivo, the induced DPCs produced fiber structures with a hair cuticle-like coat resembling the hair shaft [189]. Alternatively, scientists have directly reprogramed fibroblasts into DP-like cells using small molecules [190,191].

## 8. Conclusions and Perspectives

DPCs serve as the signaling centers for hair follicles, playing a crucial role in hair follicle development and the hair growth cycle. With recent technological advancements, DPCs have garnered increasing attention from scientists, and growing studies are exploring the functions of different genes in DPCs. It is noteworthy that the expression of different genes in DPCs undergoes dynamic changes during the hair cycle, likely involving secretory factors that promote and maintain various phases of hair growth. Hair growth is a finely tuned process in which the coordination of various secretory factors by DPCs determines the growth patterns, including initiation, termination, and pigmentation. Collecting DPCs at different time points and analyzing their expression profiles to reveal the sequential changes in DP genes and their interactions will greatly aid in understanding the hair growth signaling network. This knowledge will also establish a solid theoretical foundation for studying the mechanisms of pathological hair loss or graying.

As a unique type of MSC, DPCs can also serve as a cell source for cell therapy in various diseases, especially skin disorders like alopecia. Overcoming the barriers to DPC-based cell therapy involves maintaining stemness, while increasing passage numbers in 2D culturing, achieving cost-effective and simplified expansion in 3D culturing, and enhancing the long-term survival and functional performance of DPCs after subcutaneous injections. Continuously obtaining DPCs through iPSC induction represents a viable new approach, but improving the efficiency of iPSC induction and fully establishing this technological route will require further dedicated work from scientists.

## Figures and Tables

**Figure 1 biology-13-00842-f001:**
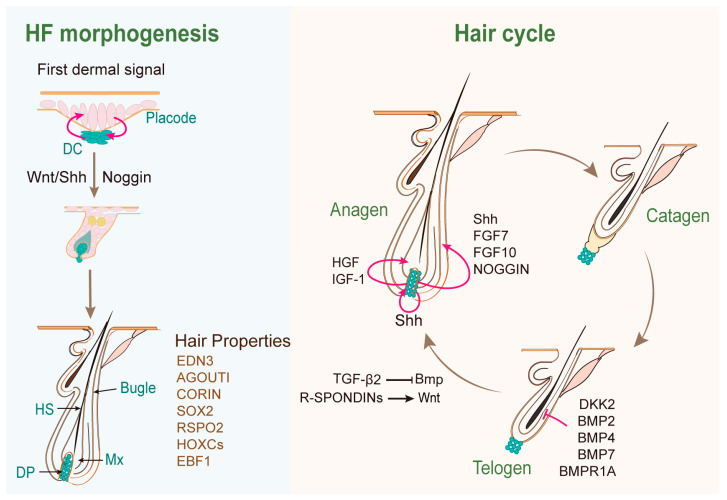
Key signaling molecules/pathways involved in the regulation of hair follicle morphogenesis and the hair cycle.

**Figure 2 biology-13-00842-f002:**
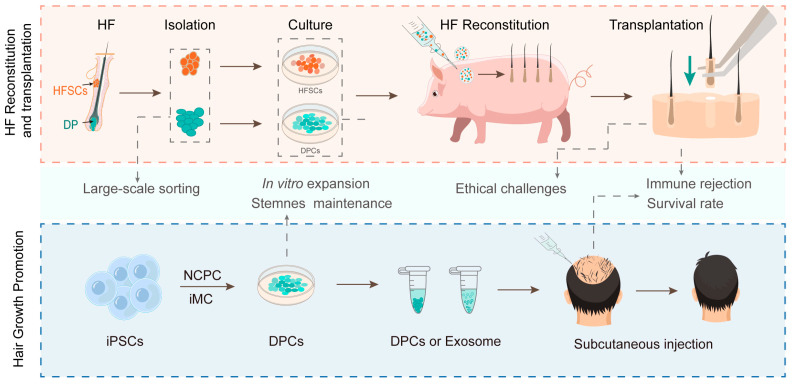
Technical and ethical challenges of two potential approaches of DPC-based cell therapy for treating alopecia. HF, hair follicle; iMC, induced mesenchymal cells. Created with BioRender.com.

**Figure 3 biology-13-00842-f003:**
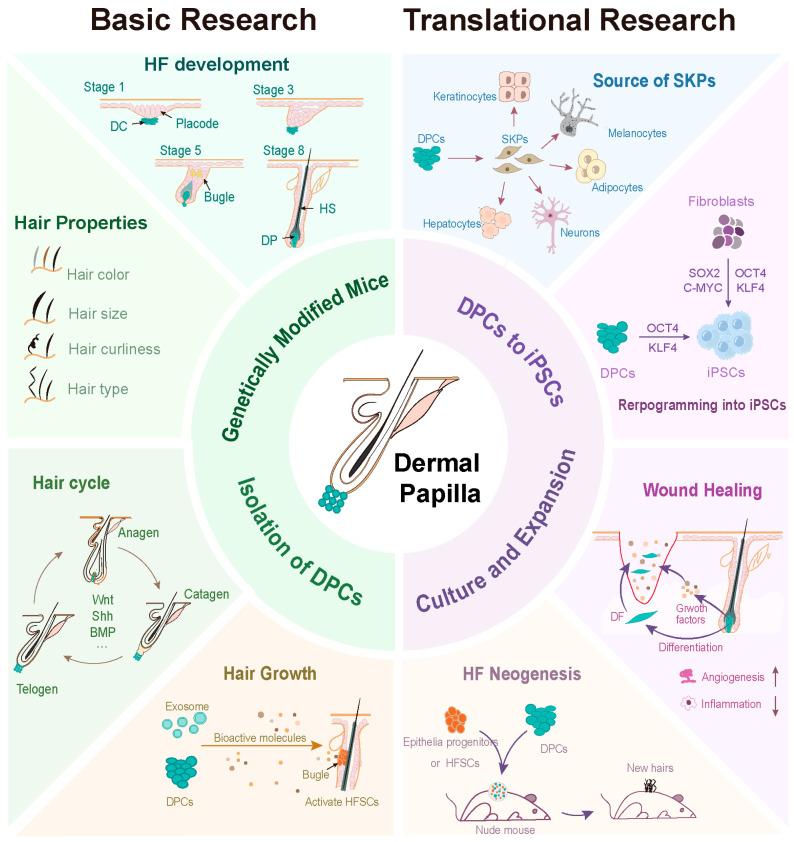
New technological advancements in basic and translational research of DPCs. HF, hair follicle; HS, hair shaft; DF, dermal fibroblasts. Created with BioRender.com (accessed on 6 August 2024).

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
