# Peer review of "Dermal Papilla Cells: From Basic Research to Translational Applications"

_biology, 2024, doi:10.3390/biology13100842_

Round 1
Reviewer 1 Report
Comments and Suggestions for Authors
The review titled Dermal Papilla Cells: from Basic Research to Translational Applications is well-written in fluent English. The authors provided a comprehensive overview of the current basic research state on dermal papilla cells, as well as its translational applications.
The review titled "Dermal Papilla Cells: From Basic Research to Translational Applications" is well-written in fluent English. This review provided a comprehensive overview of the current state of basic research on dermal papilla cells (DPCs) in hair follicle biology, as well as their potential for translational applications. By summarizing recent advancements in the field, the review contributes valuable insights, particularly regarding the therapeutic potential of DPCs and their exosomes in promoting hair growth.
The discussion of DPCs as a source of skin-derived precursors (SKPs) and their ability to induce the formation of new hair follicles is particularly compelling. The authors also underscore the unique stem cell properties of DPCs, including their ability to be reprogrammed into induced pluripotent stem cells (iPSCs) with minimal Yamanaka factors, which presents exciting possibilities for regenerative medicine.
While the review is well-structured and informative, I suggest the following improvements for clarity and depth:
1. Expansion on mechanisms: The manuscript would benefit from a deeper discussion of specific signaling pathways or transcription factors involved in these processes.
2. Challenges in translational applications: This review would be further improved by discussing on the key obstacles in moving from basic research to clinical applications, for example, the immune rejection, scalability, and long-term viability of DPC-based therapies, as well as any strategies being developed to overcome these barriers.
3. Comparative analysis: Including a comparison of DPC-based therapies with other emerging technologies for hair regeneration could provide a broader context for readers and underscore the advantages for limitations of DPC approaches.
Overall, the manuscript summarized key developments in the biology of DPCs and their potential applications in hair regeneration.
Author Response
Response to Reviewer 1 Comments |
Thank you very much for taking the time to review this manuscript. Please find the detailed point-by-point responses below and the corresponding revisions/corrections highlighted/in track changes in the re-submitted files.
Comments 1: Expansion on mechanisms: The manuscript would benefit from a deeper discussion of specific signaling pathways or transcription factors involved in these processes. Response 1: Thank you for the suggestion. We have added more detailed description of the signaling pathways or transcription factors involved in these processes (Line68-92; Line 139-155; Line 225-229, transcription factors SOX2 and EBF1 have been descripted elsewhere in the manuscript).
Comments 2: Challenges in translational applications: This review would be further improved by discussing on the key obstacles in moving from basic research to clinical applications, for example, the immune rejection, scalability, and long-term viability of DPC-based therapies, as well as any strategies being developed to overcome these barriers. Response 2: Thank you for the suggestion. We have added the description of the key obstacles to DPCs-based therapy and the strategies being developed to overcome these obstacles. To address immune rejection and long-term viability of DPCs, we describe as follows: “A common limitation of cell therapy is the poor survival of transplanted cells due to immune rejection of allogeneic cells and failure to integrate the transplanted cells into the recipient tissues/organs. To reduce immune rejection, deletion of MHC expression in trans-planted cells, pretreatment with immunosuppressive drugs are common measures. Strategies to reduce apoptotic signaling and increase cell adhesion have been developed to improve long-term survival of transplanted cells, such as pretreatment with cytokines, growth factors, and anti-apoptotic molecules. These pretreatments can likely be applied topically to the scalp when using DPCs to treat alopecia.” We have placed most parts of the technical advances after describing the challenges in translational applications. All of the technical advances, including the large scale isolation of DPCs using surface protein antibodies, the improvement of culture conditions to allow massive expansion of DPCs in vitro, and the technique of inducing iPSCs into DPCs, address the issue of scalability of DPCs.
Comments 3: Comparative analysis: Including a comparison of DPC-based therapies with other emerging technologies for hair regeneration could provide a broader context for readers and underscore the advantages for limitations of DPC approaches. Response 3: Thank you for the suggestion. We have included a comparison of DPC-based therapies with other emerging technologies, described as follows: “In addition to traditional drug therapies and hair transplantation, new therapies have been developed for the treatment of alopecia. For example, platelet-rich plasma (PRP) , MSCs, and MSCs-derived exosomes or conditioned media are injected into the scalp to stimulate HFSCs. A common mechanism behind these therapies is that the injected substances contain growth factors. DPCs are unique in that they accumulate high concentrations of growth factors that activate HFSCs, such as R-SPONDINs, VEGF, FGF, HGF, etc. DPCs-based cell therapy would offer significant ad-vantages over other therapies for the treatment of alopecia.” |
Reviewer 2 Report
Comments and Suggestions for Authors
The submitted review article, "Dermal Papilla Cells: From Basic Research to Translational Applications," provides a comprehensive summary of findings related to signaling pathways, the regulation of hair growth, potential clinical applications, and the technical challenges associated with these areas of research.
The various topics are explained in insufficient detail to provide insight into the subject matter. Furthermore, there is no evident connection between the different sections. To enhance the article, it would be beneficial to provide more detail on the mechanisms involved (especially an explanation of the interaction and importance of the signaling pathways, including illustration through figures), and discuss interspecies differences and their relevance for clinical applications.
Author Response
Response to Reviewer 2 Comments |
Thank you very much for taking the time to review this manuscript. Please find the detailed point-by-point responses below and the corresponding revisions/corrections highlighted/in track changes in the re-submitted files. Comments 1: The various topics are explained in insufficient detail to provide insight into the subject matter. Response 1: Thank you for the comments. As we reply to the reviewer 1, we have added more detailed description of the signaling pathways or transcription factors involved in functions of DPCs, more discussion on the key obstacles and strategies being developed to overcome these obstacles in moving from basic research to clinical applications, including the immune rejection, scalability, and long-term viability of DPCs, and comparative analysis of DPCs-based therapies with other emerging technologies for treating alopecia. We hope adding these details can provide more information to the subject. In this review, except summarize the current knowledge, we summarize the most important recent technological advances in DP research. 1) Establishment of Cre mice strains that can specifically knock out genes in DP. 2) Large-scale isolation of DPCs using antibodies of surface protein. 3) The improvement of culture conditions to allow massive expansion of DPCs in vitro. 4) The technique of inducing iPSCs into DPCs, providing an unlimited source of DPCs. We clearly present two potential routes of using DPCs to treat alopecia: 1) Using DPCs together with epithelial stem cells to reconstitute hair follicles in immunodeficient animals (e.g., pigs) and transplanting them to human scalps. 2) Injecting DPCs or their derived exosomes to promote hair growth. We clarify the technical and ethical challenges of these two routes of DPCs-based cell therapy for treating alopecia. We also point out some important directions for future research. For example, collecting DPCs at different time points and analyzing their expression profiles to reveal the sequential changes in DP genes and their interactions will greatly aid in understanding the signaling network of hair growth. Inducing iPSCs into neural crest stem cells (NCSCs) or induced mesenchymal cells (iMCs), and then further differentiating into DPCs represents a viable new approach for continuously obtaining enormous DPCs for cell therapy. We sincerely hope that our efforts can help colleagues in our field.
Comments 2: Furthermore, there is no evident connection between the different sections. Response 2: Thank you for your comments. We have divided the technical advances section into two parts, for basic research and for translational research, and placed them in different places to improve the connection between different sections. Our current logic and structure of the manuscript is as follows: Basic Research Findings on DP Functions in Hair Follicles (DP Regulation in Hair Follicle Development, DP Regulation in Mature Hair Follicles)- Technological Advances to Promote Basic Research on DP- Applications of DPCs in Cell Therapy- DPCs-Based Cell Therapy for the Treatment of Alopecia (Highlight)- Technological Advances to Address the Challenge of DPCs-Based Cell Therapy- Conclusions and Perspectives. We have added linking phrases to improve the logical connection between different sections. For example, to move from the description of basic research to translational application, we added the following sentence: "As the signaling center for regulating hair follicle development and hair growth, DPCs express high levels of numerous growth factors. DPCs are particularly characterized by their stem cell properties. This unique property makes them very promising for the development of cell products for cell therapy. After introducing "Stemness and Potential Applications of DPCs in Cell Therapy", we added a phase "Based on the functions in inducing hair follicle neogenesis and promoting hair growth, DPCs can be applied in two potential strategies to treat alopecia" and then introduced "DPCs-Based Cell Therapy for Treating Alopecia". After introducing the section "Strategies and Challenges of DPCs-Based Cell Therapy for Treating Alopecia" and before moving on to the section "Technological Advances in DP Translational Research- Obtaining DPCs on a Large Scale", we added a phase as follows: "Obtaining sufficient numbers of DPCs is the main obstacle to DPC-based cell therapy. To solve this problem, scientists have made significant technological advances". We hope our revision have enhanced connection between the different sections.
Comments 3: To enhance the article, it would be beneficial to provide more detail on the mechanisms involved (especially an explanation of the interaction and importance of the signaling pathways, including illustration through figures), and discuss interspecies differences and their relevance for clinical applications. Response 3: Thanks for the suggestion. We have added more details and interactions of the signaling pathways (Line68-92; Line 139-155). The species information was added when we summarized the previous reports. Most of the basic research findings are from mice, unless the species is specifically indicated. We believe that human and mouse share very similar mechanism of hair development and growth. We have added the following description of the difference in hair cycle between humans and mice: "It should be noted that in mice, the growth cycle of all hairs on the back is synchronized until the second anagen, whereas in the adult human scalp, individual hairs maintain their own growth cycle. At any given time, approximately 90% of human scalp hair follicles are in the anagen phase, 1% are in the catagen phase, and 9% are in the telogen phase [46].” |
Reviewer 3 Report
Comments and Suggestions for Authors
The present review by Zhang et al. summarizes the current literature on dermal papilla cells and discusses potential translational implications as well as challenges in the respective research area. Overall, the article is well written and a reasonable contribution to the field. The figures are well organized and help the reader understand the content of the review. However, I would like to point out some aspects in order to improve the manuscript in the interest of the reader.
1. The embryogenesis of the dermis with its various developmental origins as well as the embryology of the melanocytes as neural crest derivatives should be explained in more detail, citing relevant literature. Genes that act more downstream such as Dermo-1 could be mentioned here. Also, the role Shh should be described better.
2. At some parts, longer paragraphs are written without citing literature. For example at the beginning of 3.1. The authors should make sure to always cite appropriate literature after every statement.
3. It is not always clear, which animal model is being described, when mentioning various in vivo models in one paragraph (see line 177-179). For every statement, the animal model should be named clearly.
4. The authors mention their own more efficient technology for sorting DPCs (line 286-288). The method should be briefly described here and put into context with other pre existing methods in order to validate this statement.
5. All abbreviations have to be written out in full at least once. For example Sonic hedgehog for Shh.
To conclude, this manuscript is a reasonable contribution to the field, however major revisions should be conducted in order to be accepted for Biology.
Author Response
Response to Reviewer 3 Comments
|
Thank you very much for taking the time to review this manuscript. Please find the detailed point-by-point responses below and the corresponding revisions/corrections highlighted/in track changes in the re-submitted files. Comments 1: The embryogenesis of the dermis with its various developmental origins as well as the embryology of the melanocytes as neural crest derivatives should be explained in more detail, citing relevant literature. Genes that act more downstream such as Dermo-1 could be mentioned here. Also, the role Shh should be described better. Response 1: Thank you for the comments and suggestions. We have added a paragraph to describe the embryogenesis of the dermis with its various developmental origins, the embryology of the melanocytes as neural crest derivatives and the function of Dermo-1 as follows: “It is noteworthy that dermal fibroblasts, and thus DP, originate from different pre-cursor cells in different parts of the body. Lineage tracing assays in mice show that dermal fibroblasts in the dorsal and ventral trunk originate from the somitic and lateral plate dermomyotomes, respectively, whereas dermal fibroblasts in the head and face originate from the neural crest [30, 31]. The neural crest cells in birds and mice also migrate to the skin and give rise to melanocytes during embryogensis [31, 32]. Ablation of the Wnt sig-naling in mice dermis leads to ectopic cartilage formation in craniofacial and ventral trunk regions at the expense of dermal and bone lineages [33]. Dermo-1/twist2 (Twist Basic Helix-Loop-Helix Transcription Factor 2), a downstream transcription factor of Wnt signaling, may mediate the functionion of Wnt signaling in dermal [33, 34]. Dermo-1 knockout mice showed dramatically thin skin and sparse hair [35]. Forced expresseion of chicken Dermo-1 induces dense dermis, feathers, and scales [36]. These findings highlight the significance of Dermo-1 in dermis maturation from different embryonic origins, which is crucial for hair follicle development and patterning.” We have also expanded the description of Shh signaling during hair follicle development as follows: “The Sonic hedgehog (Shh) pathway which acts downstream of the Wnt signaling, is expressed in both epithelial cells and DC, playing a crucial role in signaling from the epithelium to both epithelial and mesenchymal cells [20]. Shh promotes the proliferation of HF epithelium and its subsequent downward growth, as well as the formation of the DC [21]. While the induction of HFs is independent of Shh, its role in the ingrowth of the epi-dermis and the subsequent morphogenesis of the hair shaft is essential [22, 23]. Shh signaling regulates specific DP signatures and maintains a reciprocal Shh–Noggin(BMP inhibitor) signaling loop to drive hair follicle morphogenesis[24].”
Comments 2: At some parts, longer paragraphs are written without citing literature. For example at the beginning of 3.1. The authors should make sure to always cite appropriate literature after every statement. Response 2: Thank you for the suggestion. We have added related literatures to support our every statement.
Comments 3: It is not always clear, which animal model is being described, when mentioning various in vivo models in one paragraph (see line 177-179). For every statement, the animal model should be named clearly. Response 3: Thank you for your suggestion. We have added the animal model information. Most of the statement is based on mouse data, but indeed some of the data came from other animal models. Comments 4: The authors mention their own more efficient technology for sorting DPCs (line 286-288). The method should be briefly described here and put into context with other pre existing methods in order to validate this statement. Response 4: Thank you for your suggestion. We have described the method in more detail and compared it with the existing methods. The full description is as follows: “Our laboratory has identified a DP membrane protein, LEPTIN Receptor (LEPR)LEPR, and developed a more specific and efficient technology for functional DPC sorting on a large scale [152]. By analyzing the gene expression profile of DP cells using existing microarray and RNA-seq databases, LEPR was identified as the second most abundant surface protein in DP cells and exhibits a higher level of specificity compared to the previously utilized CD133 marker [52, 152, 153]. The LEPR antibody-based sorting method allows the isolation of hundreds of thousands of DP cells in a few hours, a significant improvement in efficiency over traditional microdissection. Furthermore, LEPR+ cells sorted by this method retain their DP characteristics in vitro, as demonstrated by the expression of DP markers (such as alkaline phosphatase) and key secretory factors (such as RSPO1/2 and EDN3), as well as the successful reconstitution of hair follicles in nude mice when combined with epithelial stem cells [152].”
Comments 5: All abbreviations have to be written out in full at least once. For example Sonic hedgehog for Shh. Response 5: Thank you for your suggestion. We have added full names for all abbreviations. |
Round 2
Reviewer 2 Report
Comments and Suggestions for Authors
Thank you for addressing my comments. There is only one suggestion to improve the understanding of the regulation by including a figure illustrating the signaling pathways.
Author Response
Response to Reviewer Comments |
Thank you very much for taking the time to review this manuscript. Please find the detailed response below and the corresponding revisions/corrections highlighted/in track changes in the re-submitted files.
Comments 1: There is only one suggestion to improve the understanding of the regulation by including a figure illustrating the signaling pathways. Response 1: Thank you for the suggestion. We have added a figure to illustrate the signal pathways. It is worth noting that signalling pathways such as WNT, SHH and BMP are involved at different times , in different cells, during hair follicle development and hair growth, and that these signalling pathways also interact with each other. Our diagram provides a simplified treatment of this complexity. Similar descriptions can be found in other reviews. |
Reviewer 3 Report
Comments and Suggestions for Authors
Overall the changes have been well implemented. I just want to point out that there is no such thing as a lateral plate dermomyotome. The dermomyotome is a derivative of the somite. This needs to be corrected. Also in line 102 the sentence ends abruptly. There seems to be a missing word after "dermal". After this minor revision, I think the manuscript could be accepted.
Comments on the Quality of English LanguageThe English is fine. Minor corrections could be performed.
Author Response
Response to Reviewer Comments
|
Thank you very much for taking the time to review this manuscript. Please find the detailed responses below and the corresponding revisions/corrections highlighted/in track changes in the re-submitted files.
Comments 1: I just want to point out that there is no such thing as a lateral plate dermomyotome. The dermomyotome is a derivative of the somite. This needs to be corrected. Response 1: Thank you for pointing out this error. We have corrected the sentence to read as follows: “Lineage tracing assays in mice show that dermal fibroblasts in the dorsal and ventral trunk originate from the somite, whereas dermal fibroblasts in the head and face originate from the neural crest.” |
Comments 2: Also in line 102 the sentence ends abruptly. There seems to be a missing word after "dermal". After this minor revision, I think the manuscript could be accepted.
Response 2: Thank you for pointing out this error. We have corrected the sentence to read as follows: ”Dermo-1/twist2 (Twist Basic Helix-Loop-Helix Transcription Factor 2), a downstream transcription factor of WNT signaling, may mediate the function of WNT signaling in dermal cell development.”